# The aetiology and clinical characteristics of cryptococcal infections in Far North Queensland, tropical Australia

**Beatrice Z. Sim**[1,2], **Luke Conway**[1], **Laura K. Smith**[1], **Lee Fairhead**[1], **Yi Shan Der**[1], **Lara Payne**[1], **Enzo Binotto**[1], **Simon Smith**[1], **Josh Hanson**[1,3] *

**1** Department of Medicine, Cairns Hospital, Cairns, Australia, **2** School of Medicine, University of Queensland, Brisbane, Australia, **3** Kirby Institute, University of New South Wales, Sydney, Australia

* jhanson@kirby.unsw.edu.au

**Data Availability Statement:** All relevant data are within the paper and its Supporting Information files.

## Abstract

Cryptococcal infections are an important cause of morbidity and mortality in tropical Australia. This retrospective audit was conducted to characterise the aetiology, temporospatial epidemiology, and clinical course of 49 cryptococcal infections in Far North Queensland between 1 January 1999 and 31 December 2019. *Cryptococcus gattii* was identified in 15/32 (47%) in whom it was possible to speciate the organism. Among these 15 patients, 13 (87%) had a rural residential address, 10 (67%) were Indigenous Australians and 11 (73%) presented during the May-November dry season. When compared to the 17 patients with *Cryptococcus neoformans* infection, patients with *C. gattii* were less likely to be immunocompromised (0/15 versus 8/17 (47%), p = 0.003). Neurosurgery was necessary in 5/15 *C. gattii* cases and 3/17 (18%) *C. neoformans* cases (p = 0.42). Outcomes were generally good with 42/49 (86%) cases—and 14/15 (93%) with *C. gattii* infection—surviving to hospital discharge. These positive outcomes are likely to be explained by the development of standardised treatment guidelines during the study period, low rates of comorbidity in the patients with *C. gattii* infection and access to liposomal amphotericin and neurosurgical support in the well-resourced Australian healthcare system.

## Introduction

*Cryptococcus gattii* and *Cryptococcus neoformans* are important fungal pathogens which usually affect the central nervous system (CNS) or lungs [1]. However, there are important differences between the two infections. Globally, *C. neoformans* is a more common cause of disease, predominantly affecting individuals with impaired cell-mediated immunity, particularly those with human immunodeficiency virus (HIV) or transplant recipients [1, 2]. *C. gattii*, by contrast, commonly presents with large cryptococcomas in patients with no apparent immunocompromise [3]. In the CNS, these cryptococcomas can necessitate urgent neurosurgical intervention and are associated with a poor prognosis [4, 5]. In the lung, they can be mistaken for malignancy, though presentations with pneumonia and respiratory failure have also been reported [1, 6].

**Funding:** The authors received no specific funding for this work.

**Competing interests:** The authors have declared that no competing interests exist.

*C. gattii* was responsible for only 15% of the cryptococcal infections in a prospective, community-based Australasian series performed between 1994 and 1997, although this is likely to be reflective of the HIV epidemic at that time [3]. In a more recent study from regional New South Wales, it accounted for 28/107 (26%) cases between 2003 and 2016, while in a study from an urban centre in sub-tropical Queensland between 2001 and 2015, it accounted for 14/97 (14%) infections [7, 8].

In tropical Australia, the proportion of cryptococcal infections caused by *C. gattii* is higher. In a series from the tropical Northern Territory, *C. gattii* was responsible for 12/18 (67%) cases of cryptococcal disease between 1993 and 2000 [9]. The authors identified that *C. gattii* infections occurred more commonly in Aboriginal Australians living in remote locations, a finding that might be explained by their proximity to the *Eucalyptus camaldulensis* trees that—with *E. tereticornis*—have long been recognised as an environmental niche for *C. gattii* in Australia [10, 11]. In this Northern Territory series 13/18 (72%) patients had pulmonary infections, of whom, only 6 (46%) had concurrent CNS disease [9]. The patients had significant morbidity and mortality: 8/18 (44%) required surgery for associated complications, and 4/18 (22%) died [9].

*E. camaldulensis* and *E. tereticornis* are abundant in Far North Queensland (FNQ) in tropical Australia (11) (Fig 1). Anecdotally, there is also a significant local burden of cryptococcal infection, however the microbiology, epidemiology, presentation, and clinical course of these infections has been incompletely described. This study aimed to gain a greater understanding of the characteristics and outcomes of cryptococcal infections in tropical Australia during a period which saw enormous progress in HIV care and the development of novel—and better tolerated—antifungal therapies.

## Methods

### Data collection

This retrospective study was performed at Cairns Hospital, a 531-bed tertiary referral hospital that serves a population of approximately 280,000 people who live across an area of 380,748 km$^2$ (Fig 1); 17% of the general local population identify as Indigenous Australians [12]. Patients admitted to Cairns Hospital with a laboratory-confirmed diagnosis of cryptococcal infection between 1 January 1999 and 31 December 2019 were eligible for inclusion into the study. Laboratory confirmed infection was defined as a compatible clinical syndrome with a positive culture for Cryptococcus, a cryptococcal antigen (CrAg) detected on serum or cerebrospinal fluid, or tissue histology consistent with cryptococcosis. The study period was selected as it coincided with the introduction of a statewide electronic pathology system, AUSLAB.

Patients were identified by performing a search in AUSLAB for all laboratory confirmed cases within the study period. Once identified, the patients' medical records were reviewed and their demographic, clinical, laboratory, and microbiological results were recorded.

Patients were defined as living in a rural/remote, or urban location using the Australian Standard Geographical Classification–Remoteness Areas (ASGC-RA) classification [13]. All individuals receiving care in Queensland's public health system, are asked whether they identify as an Aboriginal Australian, a Torres Strait Islander Australian, both, or neither. Comorbidity was quantified using the Charlson comorbidity index and determined using the patients' medical record [14]. If individual comorbidities were not documented, they were presumed to be absent. Patients were defined as immunocompromised if they were receiving any immunosuppressive or immunomodulatory therapy (including any dose of systemic corticosteroids), were living with HIV or had an active malignancy. Hazardous alcohol use was

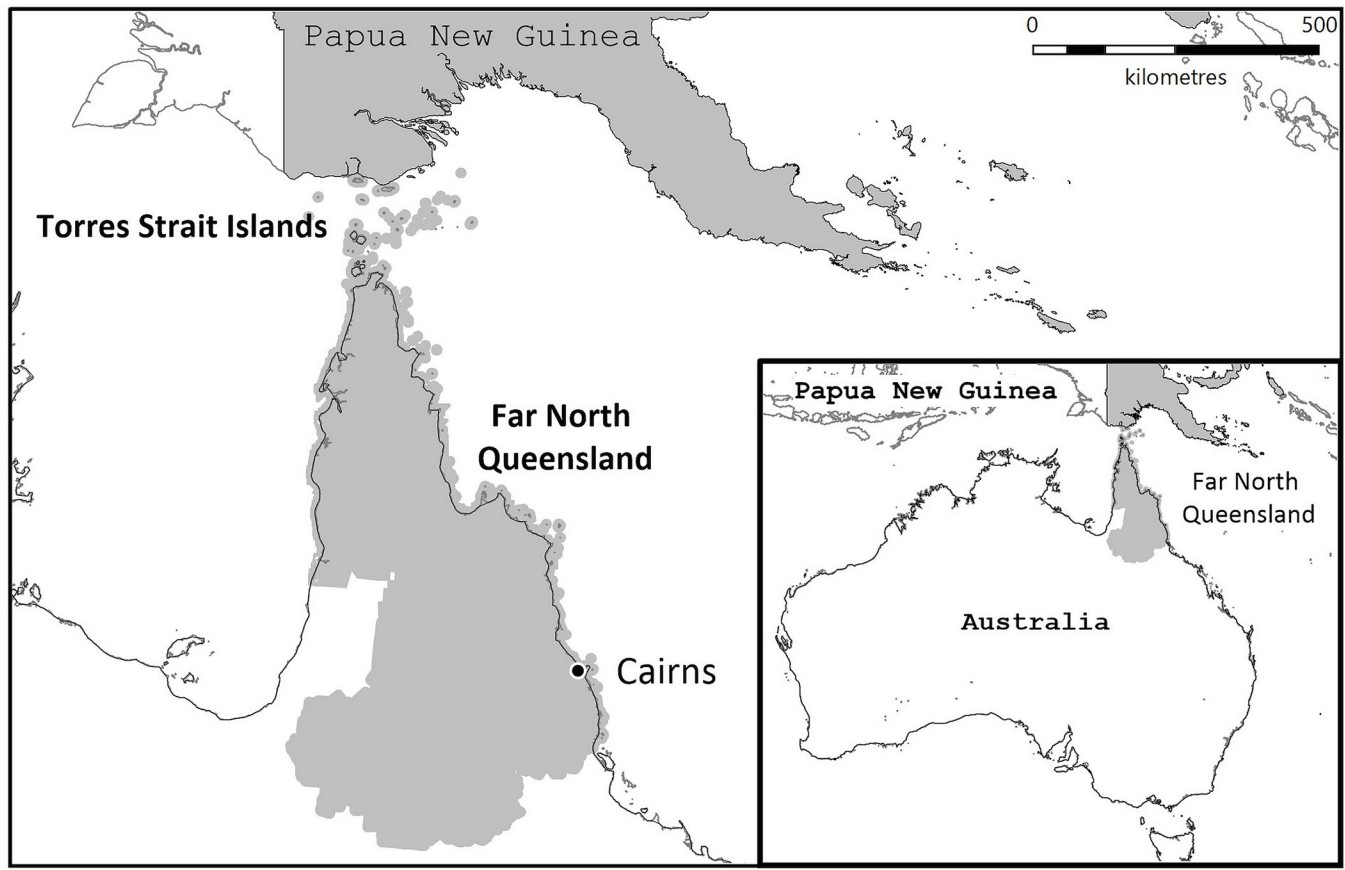

**Fig 1. The study region.** The map was created using constructed using mapping software (MapInfo version 15.02, Connecticut, USA) using data provided by the State of Queensland (QSpatial). Queensland Place Names—State of Queensland (Department of Natural Resources, Mines and Energy) 2019, available under Creative Commons Attribution 4.0 International licence https://creativecommons.org/licenses/by/4.0/. 'Coastline and state border–Queensland—State of Queensland (Department of Natural Resources, Mines and Energy) 2019, available under Creative Commons Attribution 4.0 International licence https://creativecommons.org/licenses/by/4.0/.

defined as more than 20 grams of alcohol per day as per 2009 Australian guidelines [15]. A cryptococcoma was defined as a lesion (on either lung or brain imaging) ≥ 10mm in size.

Local flowering times of *E. tereticornis* and *E. camaldulensis* were determined using data from the Australasian Virtual Herbarium [16]. Australian Bureau of Statistics population data were used to calculate disease incidence [17].

## Statistical analysis

Data was de-identified, entered into an electronic database (Microsoft Excel 2016, Microsoft, Redmond, WA, USA) and analysed using statistical software (Stata version 14.2, StataCorp LLC, College Station, TX, USA). Univariate analysis was performed using the Kruskal-Wallis, Chi-squared, Spearman's rho or Fisher's exact tests where appropriate. Trends over time with determined using an extension of the Wilcoxon rank-sum test with year as a continuous variable [18].

## Ethics approval

The FNQ Human Research Ethics Committee provided ethical approval for the study (HREC/16/QCH/110–1083 LR). As the data were retrospective and de-identified, the Committee waived the requirement for informed consent.

## Results

A total of 49 patients were identified and included in the analysis (S1 Fig). The diagnosis of cryptococcal infection was made by culture in 19/49 (39%), serum CrAg in 18/49 (37%), CSF CrAg in 6/49 (12.2%) and histologically in 6/49 (including 1 at autopsy). Speciation was possible in 32/49 (65%): 17/32 (53%) were *C. neoformans* and 15/32 (47%) were *C. gattii*.

The median (interquartile range (IQR)) age of patients in the cohort was 48 (37–58) years; 30 (61%) were male; 16 (33%) were immunocompromised. Patients with *C. neoformans* infection were more likely to be immunosuppressed (47% vs 0%, p = 0.003) while patients with *C. gattii* were more likely to be smokers (73% vs 24%, p = 0.01). Differences in the other characteristics of the *C. neoformans* and *C. gattii* cases are presented in Table 1.

### Indigenous Australians

Overall, 27/49 (55%) identified as either an Aboriginal or a Torres Strait Islander Australian compared to 49241/287107 (17.2%) of the general FNQ population at the end of the study period (p<0.0001). Indigenous patients were less likely to be immunocompromised than non-Indigenous patients (4/27 (15%) versus 12/22 (55%), p = 0.005); 10/15 (67%) *C. gattii* cases and 8/17 (47%) *C. neoformans* cases occurred in Indigenous Australians (p = 0.31). There was no difference between Indigenous and non-Indigenous Australians in age, gender or residence in a remote location. However, Indigenous Australians were more likely to smoke tobacco (17/27 (63%) versus 6/22 (27%), p = 0.02) or have hazardous alcohol use (16/27 (59%) versus 2/22 (9%), p<0.0001). There was no difference in clinical course and outcome (Table 2).

### Temporospatial epidemiology

The absolute number of cases of cryptococcal infection that were diagnosed increased over the study period (p for trend = 0.03), although there was no change in the annual incidence per 1,000,000 population (p for trend = 0.89) (Fig 2). In the final year of the study period, the 4 cases of cryptococcal disease—all *C. gattii*—represent an annual local incidence of 14.3 cases

**Table 1. Comparison of selected demographic and clinical characteristics of the patients with confirmed *C. gattii* and *C. neoformans* infection.**

| | Unspeciated n = 17 | *C. gattii* n = 15 | *C. neoformans* n = 17 | P [a] |
|---|---|---|---|---|
| Age | 52 (37–71) | 45 (18–82) | 47 (38–54) | 0.64 |
| Male gender | 13 (76%) | 8 (53%) | 9 (53%) | 1.0 |
| Indigenous Australian | 9 (53%) | 10 (67%) | 8 (47%) | 0.31 |
| Rural residential address [b] | 6 (38%) | 13 (87%) | 9/13 (69%) | 0.37 |
| Charlson Comorbidity Index = 0 | 2 (12%) | 8 (53%) | 5 (29%) | 0.28 |
| Charlson Comorbidity Index | 0 (0–0) | 0 (0–4) | 3 (0–6) | 0.13 |
| Known immunosuppression | 8 (47%) | 0 (0%) | 8 (47%) | 0.003 |
| HIV infection | 2 (12%) | 0 (0%) | 3 (18%) | 0.23 |
| Solid organ transplant recipient | 1 (6%) | 0 (0%) | 2 (12%) | 0.49 |
| Renal replacement therapy | 1 (6%) | 1 (7%) | 1 (6%) | 1.0 |
| Hazardous alcohol use (past or current) | 5 (29%) | 7 (47%) | 6 (35%) | 0.72 |
| Smoker (past or current) | 8 (47%) | 11 (73%) | 4 (24%) | 0.01 |

All data are presented as n (%) or median (interquartile range)

[a] For comparison between patients with confirmed *C. neoformans* and *C. gattii*

[b] 4 patients with confirmed *C. neoformans* and 1 patient with a non-speciated infection were not local residents of FNQ

HIV: Human immunodeficiency virus

**Table 2. Comparison of selected demographic and clinical characteristics of the patients.**

|  | Indigenous Australian n = 27 | non-Indigenous Australian n = 22 | p |
|---|---|---|---|
| Age (years) | 47 (33–53) | 49 (39–71) | 0.29 |
| Male gender | 17 (63%) | 13 (59%) | 0.78 |
| Rural/remote residence [a] | 20 (74%) | 8/17 (47%) | 0.11 |
| *C. gattii* [b] | 10/18 (56%) | 5/14 (36%) | 0.31 |
| Brain involvement | 19 (70%) | 14 (64%) | 0.76 |
| Lung involvement | 23 (85%) | 10 (45%) | 0.005 |
| HIV/AIDS | 1 (4%) | 3 (14%) | 0.31 |
| Solid organ transplant | 0 | 3 (14%) | 0.08 |
| Immunocompromised | 4 (15%) | 12 (55%) | 0.005 |
| Charlson Comorbidity Index | 2 (0–4) | 3 (0–5) | 0.34 |
| Underlying chronic lung disease | 4 (15%) | 4 (18%) | 1.0 |
| Diabetes mellitus | 8 (30%) | 3 (14%) | 0.30 |
| Receiving dialysis | 2 (7%) | 1 (5%) | 1.0 |
| Smoker (past or current) | 17 (63%) | 6 (27%) | 0.02 |
| Hazardous alcohol use (past or current) | 16 (59%) | 2 (9%) | <0.0001 |
| Lumbar puncture performed | 21 (78%) | 19 (86%) | 0.49 |
| Lumbar puncture opening pressure [c] | 19 (15–33) | 34 (17–38) | 0.16 |
| Induction Liposomal Amphotericin [d] | 11/19 (58%) | 10/17 (59%) | 1.0 |
| Neurosurgery required | 5 (19%) | 3 (14%) | 0.72 |
| Intensive Care Unit admission | 2 (7%) | 5 (23%) | 0.22 |
| Died | 5 (19%) | 2 (9%) | 0.44 |

[a] There were 5 individuals who were not local residents

[b] Only includes the 32 cases in which a species was identified

[c] Only in the patients with CNS involvement

[d] Of the 36 patients who had liposomal amphotericin

per 1,000,000 population. However, in this small sample there was no discernible change in proportion of *C. gattii* cases over time; the mean (95% confidence interval) annual incidence of *C. gattii* over the entire study period was 2.8 (1.1–4.5) per 1,000,000 population. *C. gattii* cases were observed most commonly in July, which coincides with the peak local flowering of *E. camaldulensis* (Fig 3) (16).

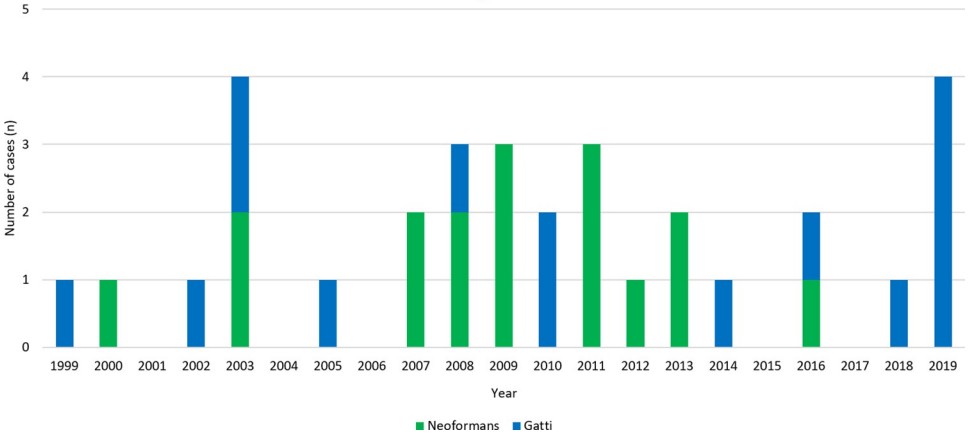

**Fig 2. The proportion of Cryptococcal infections caused by *C. gattii* during the study period.**

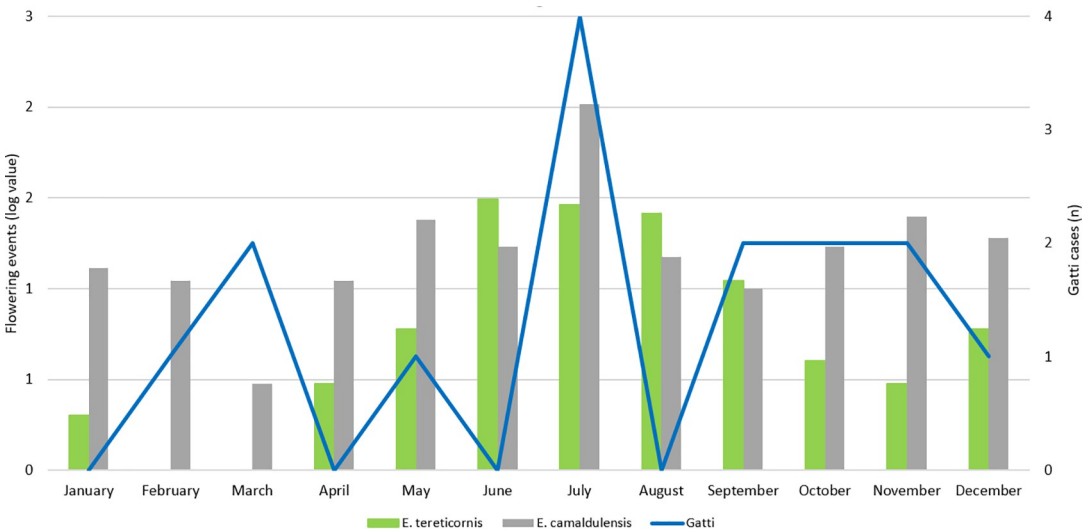

**Fig 3. Seasonal variation in the presentation of *C. gattii* cases related to the flowering times of *E. tereticornis* and *E. camaldulensis*.**

Among the 44 local residents, 27 (61%) lived in a rural/remote location, compared to 10.4% of the local general population at the end of the study period (p<0.0001). Among the remaining 5 cases, 2 were resident in Papua New Guinea, 2 lived in other states in temperate Australia and 1 was a South African resident. All 15 *C. gattii* cases occurred in local residents, 13/15 (87%) of *C. gattii* cases occurred in patients living in a rural or remote area, compared with 9/13 (69%) of *C. neoformans* cases (p = 0.20), however there was no clear temporospatial clustering (Fig 4).

### Site of infection

18/49 (36.7%) patients had concurrent pulmonary and central nervous system CNS infection (S2 Fig). Current or past smoking was more common in patients with lung involvement than without lung involvement (19/33 (58%) versus 4/16 (25%), p = 0.04). Among patients with confirmed *C. gattii* infection, 4/15 (27%) had an isolated CNS presentation, 2/15 (13%) had an isolated respiratory presentation, while in 9/15 (60%) concurrent CNS and lung involvement was confirmed. Patients with *C. neoformans* infections presented with isolated CNS presentation in 10/17 (59%), isolated respiratory infection in 1/17 (6%), isolated skin infection in 1/17 (6%) and concurrent CNS and lung infections in 5/17 (29%) (S3 Table).

### Imaging

The small number of cases in which speciation was possible precluded meaningful comparison of the imaging findings in the *C. gattii* and *C. neoformans* cases, however, the lung imaging findings were similar. It was possible to review the brain imaging in 24/33 cases of the patients with CNS involvement; 4/24 (16.7%) had cryptococcomas; all three of the cases with CNS cryptococcomas in which speciation was possible had *C. gattii* isolated. No patient had hydrocephalus at presentation (Fig 5, S2 Table).

### Laboratory investigations

There was no significant difference in blood results between the *C. neoformans* and *C. gattii* cases apart from a lower lymphocyte count in the *C. neoformans* group (0.8 (0.5–1.3) x $10^6$/L

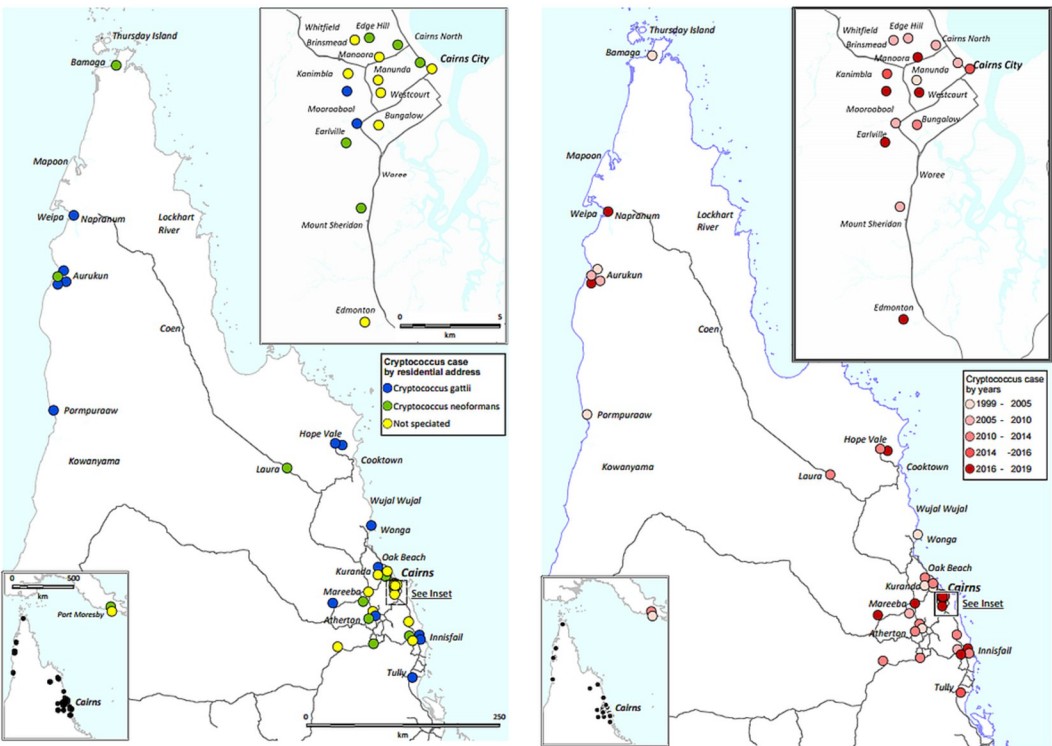

**Fig 4. Temporospatial distribution of cases.** These maps were created using constructed using mapping software (MapInfo version 15.02, Connecticut, USA) using data provided by the State of Queensland (QSpatial). Queensland Place Names—State of Queensland (Department of Natural Resources, Mines and Energy) 2019, available under Creative Commons Attribution 4.0 International licence https://creativecommons.org/licenses/by/4.0/. 'Coastline and state border–Queensland—State of Queensland (Department of Natural Resources, Mines and Energy) 2019, available under Creative Commons Attribution 4.0 International licence https://creativecommons.org/licenses/by/4.0/.

versus 1.4 (0.9–2.3), p = 0.04) (S4 Table), likely reflecting higher rates of immunocompromise in these patients. A lumbar puncture (LP) was performed in 40/49 (82%); 3 patients died before an LP could be performed, 1 declined the investigation, 1 was transferred to another hospital, in the remaining 4, the attending clinicians elected not to perform the procedure. The median (IQR) opening pressure was 27 (16–35) cm $H_2O$ in 10 confirmed *C. gattii* infections compared to 32 (16–35) cm $H_2O$ in 11 confirmed *C. neoformans* cases (p = 0.70) (S5 Table).

## Antimicrobial susceptibility

The sensitivities of the isolates are presented in Fig 6 and Table 3. The minimum inhibitory concentrations (MICs) of both organisms to the induction antifungals, amphotericin and 5-flucytosine, were comparable. The median (IQR) MIC to fluconazole was 8 (4–16) μg/mL in *C. gattii* isolates compared to 4 (2–8) μg/mL in *C. neoformans* isolates (p = 0.17).

## Antifungal therapy

45/49 patients received anti-fungal therapy; 4 (8%) died before treatment could be initiated. All 45 patients received either amphotericin B or liposomal amphotericin induction therapy. One patient with *C. gattii* and one patient with *C. neoformans* received monotherapy, but all others received adjunctive 5-flucytosine. Anti-fungal therapy doses were similar in both species: 13/15 *C. gattii* patients received induction therapy with a median (IQR) daily dose of 3.5

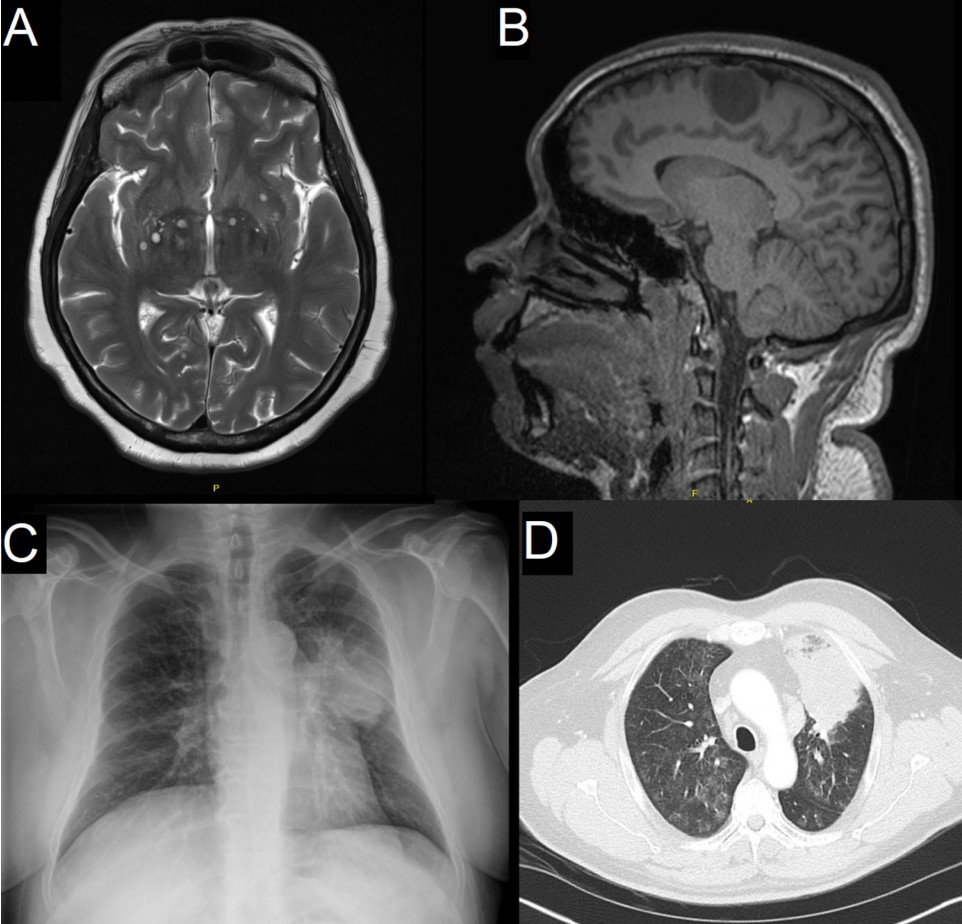

**Fig 5. Imaging findings in patients with C. gattii infection.**

(2.0–4.7) mg/kg liposomal amphotericin or 0.7 (0.6–1.0) mg/kg conventional amphotericin;15/17 *C. neoformans* patients received a median (IQR) daily dose of 3.5 (3.2–3.8) mg/kg liposomal amphotericin or 0.7 (0.6–0.9) mg/kg conventional amphotericin. The median (IQR) induction period in the patients who survived to the end of their induction was 4 (2–6) weeks in cases of *C. gattii* compared to 3 (2–6) weeks in cases of *C. neoformans* (p = 0.72). Consolidation therapy was fluconazole with a median dose of 400 (200–800) mg/day in both groups (p = 0.54).

There was no correlation between fluconazole MIC and fluconazole dose prescribed to either *C. gatii* ($r_s$ = 0.04, p = 0.90) or *C. neoformans* ($r_s$ = 0.10, p = 0.73) infections (S6 Table). Despite this, *C. gattii* patients had good outcomes. There was only one death in this group which occurred in a patient whose isolate had a fluconazole MIC was 16 μg/mL. This patient died four months after his initial diagnosis, after 8 weeks of induction amphotericin B and 5-flucytosine and following multiple admissions—and neurosurgery—for increased intracranial pressure while receiving a consolidation dose of 800mg fluconazole daily (S7 Table, Fig 7).

## Clinical course

There were 7/49 (14%) who were admitted to the intensive care unit (ICU) for neurological sequelae. Neurosurgical intervention was necessary in 8/33 (24.2%) cases with CNS

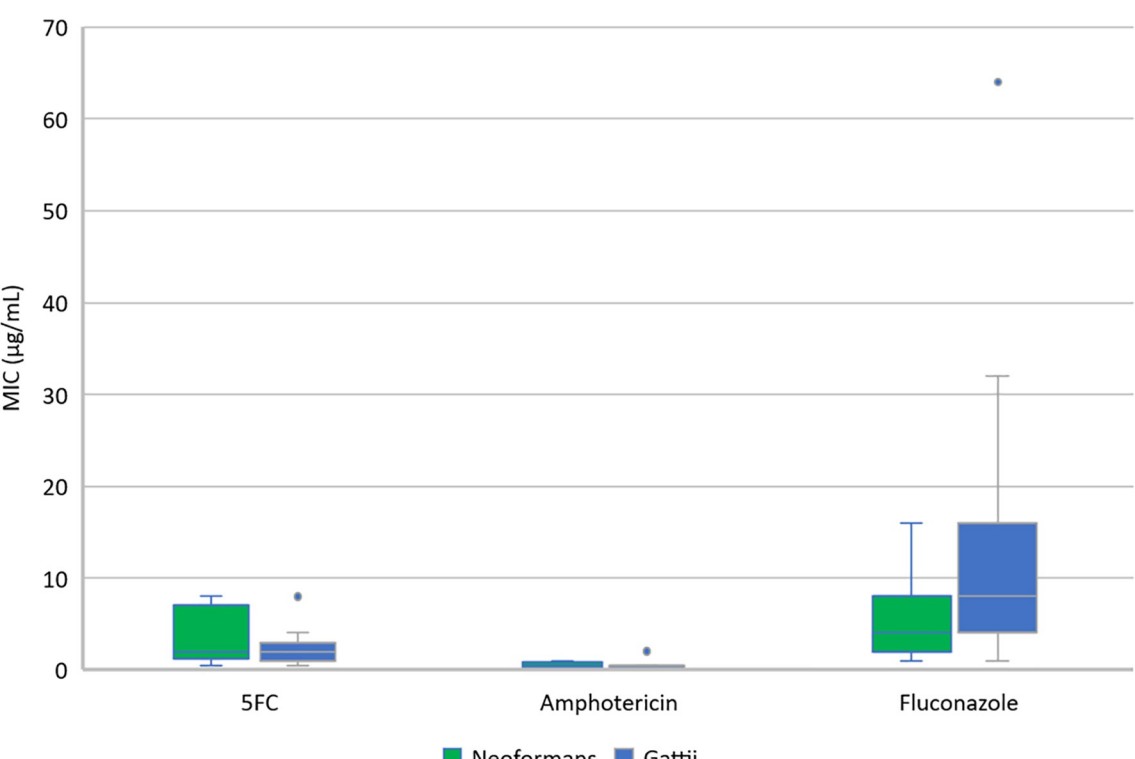

**Fig 6. Minimum Inhibitory Concentration (MIC) of *C. gattii* and *C. neoformans* isolates to commonly used antifungal agents.**

involvement; these 8 cases included 5/13 (39%) *C. gattii* and 3/15 (20%) *C. neoformans* cases (p = 0.41). Neurosurgical procedures included lumbar and extraventricular drains and ventriculoperitoneal shunting. Steroids were prescribed in 11/33 (33%) of CNS cases; 2/13 (17%) *C. gattii* and 7/15 (47%) *C. neoformans* cases (p = 0.11).

Of the 49 patients, 42 (86%) survived to hospital discharge. Twelve months after discharge, 13/42 (31%) had been lost to follow up, 17/42 (40%) were asymptomatic or improving, 6/42 (14%) had been readmitted with relapse, 4/42 (10%) had been hospitalised with an unrelated problem and 1/42 (2%) had died from the underlying lymphoma that had predisposed them to infection (Fig 7).

In this small sample there was no statistically significant association between age, Indigenous status, comorbidity, and in-hospital death. The median (IQR) age of patients that died was 48 (47–76) versus 48 (34–57) in survivors (p = 0.21); 5/27 (19%) Indigenous patients died compared with 2/22 (9%) non-Indigenous patients (p = 0.44); the median (IQR) Charlson Comorbidity index among patients that died was 2 (1–4) versus 2 (0–5) among survivors

**Table 3. Minimum Inhibitory Concentration (MIC) of *C. gattii* and *C. neoformans* isolates to commonly used antifungal agents.**

|  | 5-Flucytosine | Amphotericin | Fluconazole | Itraconazole | Ketoconazole | Voriconazole | Posaconazole |
|---|---|---|---|---|---|---|---|
| ***C. gattii*** | 2 (0.5–8) | 0.5 (0.06–2) | 8 (1–64) | 0.09 (0.015–0.5) | 0.09 (0.016–0.125) | 0.06 (0.015–0.25) | 0.12 (0.03–0.5) |
| ***Number of C. gattii isolates*** | 13 | 13 | 13 | 12 | 6 | 10 | 7 |
| ***C. neoformans*** | 2 (0.5–8) | 0.5 (0.12–1) | 4 (1–16) | 0.125 (0.015–0.25) | 0.06 (0.016–0.25) | 0.03 (0.015–0.125) | 0.12 (0.03–0.5) |
| ***Number of C. neoformans isolates*** | 16 | 16 | 16 | 14 | 14 | 14 | 11 |

All concentrations are in µg/mL

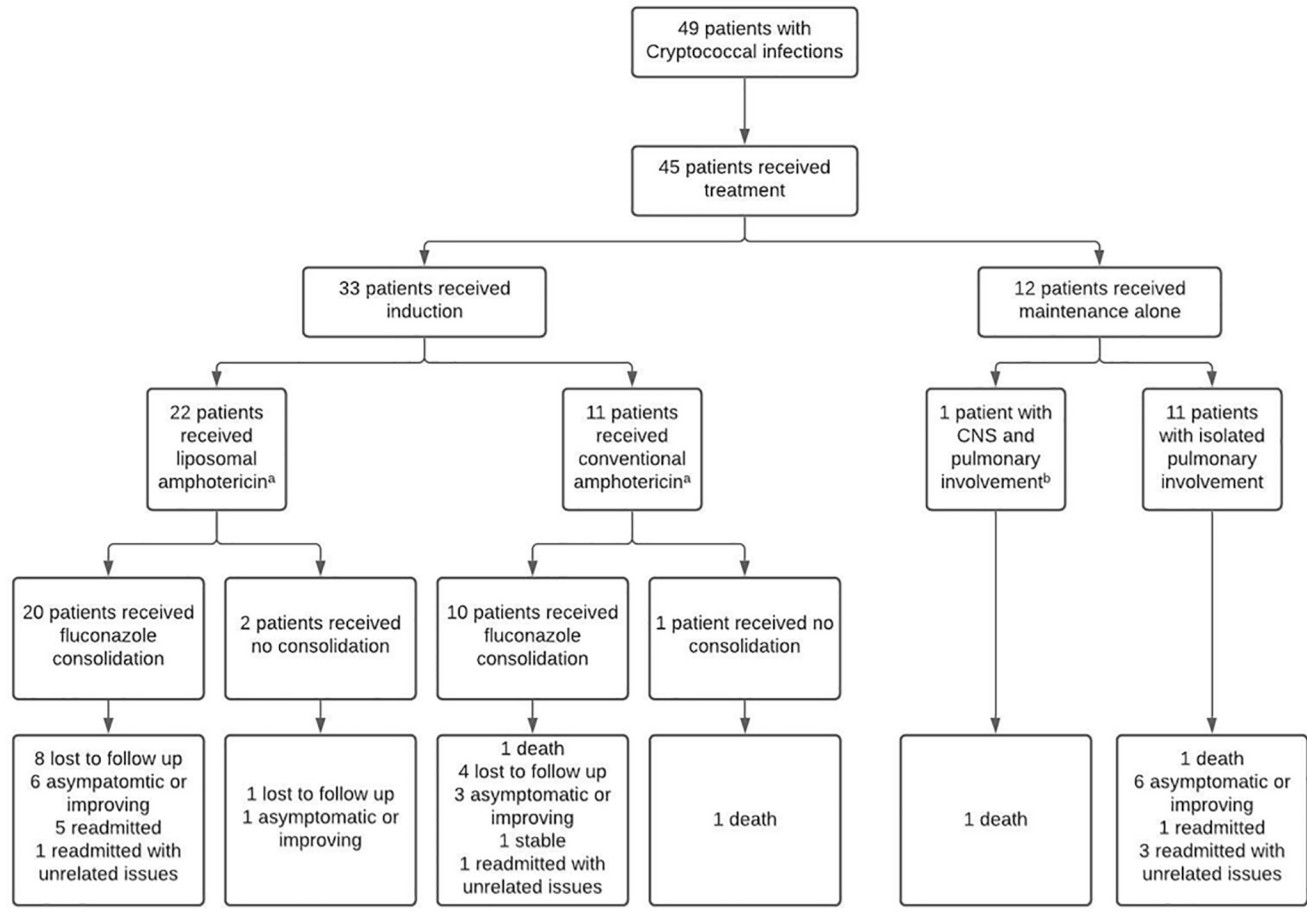

**Fig 7. Treatment course and outcomes of the patients included in the study.**

(p = 0.87); 3/16 (19%) of immunocompromised patients died versus 4/33 (12%) immunocompetent patients (p = 0.67). There was no difference in the case fatality rate of patients with and without confirmed CNS involvement (5/33 (15%) versus 2/16 (13%), p = 1.0).

It was notable that among the 33 patients who received amphotericin induction therapy, there were no deaths among the 22 patients who received liposomal amphotericin in their induction regimen (21 in combination with 5FC and one as monotherapy), whereas 2/11 (18%) who received conventional amphotericin B died (p = 0.04).

## Discussion

*C. neoformans* is the predominant cause of cryptococcal infections globally, however in tropical Australia a significant proportion are due to *C. gattii*; almost half of the speciated cryptococcal infections in this series were caused by this organism.

Australian studies from other locations have described a significantly lower prevalence of *C. gattii* infections, in keeping with their predominantly urban population [7, 8, 19]., although the Northern Territory of Australia is a notable exception [3, 9, 20]. The associations with *C. gattii* infection described in the Northern Territory—Indigenous status and rural residence—were also seen in this FNQ cohort: two-thirds of the patients with *C. gattii* in FNQ identified as an Indigenous Australian, despite their representing only 17% of the local general

population [17]. Meanwhile, almost 90% of patients with *C. gattii* in FNQ lived in rural regions compared with only 46% of the local general population.

Patients with *C. gattii* are frequently immunocompetent with no apparent risk factor for the disease [21]. In this series, every patient with confirmed *C. gattii* infection was immunocompetent, and half had a Charlson Comorbidity Index of zero. However, *C. gattii* infections were more common in current or former smokers while almost half had a history of hazardous alcohol use [22–25]. Both of these risk factors—markers of greater socioeconomic disadvantage—were more common in the Indigenous patients in this series and might be responsible for their over-representation in this cohort. Indeed, smoking and hazardous alcohol make a significant contribution to the disproportionate burden of communicable and non-communicable diseases in the local Indigenous population and are an obvious target for more aggressive preventative health strategies in the region [26, 27].

Almost 90% of the *C. gattii* infections in this cohort had CNS involvement, a similar finding to that seen in Northern Territory data where 56% of *C. gattii* cases also had CNS involvement [9]. In contrast, a large Canadian series reported that CNS involvement was present in less than 20% of their *C. gattii* cases [28]. Geographic differences in the predominant genotype may account for this: in Australia, the main genotype is VGI, whereas in Canada, it is VGIIa. However, the pathogenesis of *C. gattii* infections is incompletely defined with the role of the organism's genotype and virulence factors—and the host's genetics and immunity—yet to be fully elucidated [21, 28, 29].

Eucalypt trees act as an environmental niche for *C. gattii* in Australia and this is likely to account for the higher proportion of cases in residents of rural regions [10, 30]. *E. camaldulensis* and *E. tereticornis* flower primarily from May to September in FNQ which coincides with the July peak in *C. gattii* infections seen in this study (Fig 3) [16]. Previous air-sampling experiments in temperate South Australia identified airborne propagules of *C. gattii* under the canopy of an *E. camaldulensis* tree in flower, while all other air-sampling experiments—including those conducted under *E. camaldulensis* trees not in flower, failed to detect *C. gattii*—suggesting that dispersal of infectious propagules at the time of flowering might represent one mechanism by which *C. gattii* can infect the host [10, 31]. While hypothesis-generating and worthy of further study, it is difficult to ascribe causation to this observation given the uncertainty over the infection's incubation period [8, 32, 33]. Furthermore, in British Columbia, while *C. gattii* was identified in air samples more commonly during the warm, dry summer months, there was no association between the flowering or pollination times of individual tree species and the detected concentrations of airborne *C. gattii* [34]. It is also possible that other factors which impact on the incidence of infectious diseases in FNQ—including urban expansion, the impact of the local monsoonal wet season on vegetation growth and human activity—play a role in the temporospatial epidemiology of *C. gattii* in the region [35–40].

Antimicrobial resistance profiles of the *C. gattii* isolates in this study were similar to those reported in the literature; *C. gattii* were reliably sensitive to 5-flucytosine and amphotericin at a comparable minimum inhibitory concentration (MIC) to *C. neoformans*, while the median MIC to fluconazole was twice as high [41]. However, despite this, the dose of fluconazole prescribed by clinicians for consolidation therapy of both *C. gattii* and *C. neoformans* was similar.

Outcomes for patients with *C. gattii* were excellent with only a single death among the *C. gattii* cases (overall case fatality rate of 6.7%). In the Northern Territory series, which examined cases between 1993 and 2000, the case-fatality rate was 22%. Advances in local healthcare since the turn of the millennium including improved access to healthcare and enhanced aeromedical retrieval may also have contributed to the good outcomes [42–44]. In addition, all our patients were immunocompetent which may explain their lower case fatality rate; in one large Australian series of *C. gattii* cases, the case fatality rate was 29.2% in immunocompromised patients

compared to 4.8% immunocompetent hosts [45]. The greater use of liposomal amphotericin in this series is likely to have reduced adverse effects of therapy [46, 47]. While the efficacy of a lower dose of liposomal amphotericin has been suggested, the dose used in this cohort was within the conventional range [7].

This study was limited by its size and the significant number of infections that could not be speciated; its retrospective nature precluded the collection of comprehensive data in all patients. The patients' residential address was recorded, and this would not necessarily represent the location where the infection was contracted. The cohort included a heterogeneous group of patients: about a third had disease limited to the lung or skin, a group which traditionally has better outcomes than those with CNS disease [33], although there was no difference in outcome between patients with and without CNS disease in this small study. Some of the patients diagnosed with cryptococcal infection in the Cairns Hospital laboratory were not managed at the hospital and were therefore not included in the analysis. These factors would all tend to increase the likelihood of type 2 statistical errors. It would also tend to under-estimate the local incidence of disease. Nevertheless, it demonstrates the disproportionate burden of *C. gattii* infection in this region of tropical Australia and provides an insight into the presentation and clinical course of the infection in contemporary Australia. The association between flowering of *E. camaldulensis* and *E. tereticornis* and presentation requires further elucidation and is the focus of ongoing work. Future studies focussing on the optimal dose and duration of induction and eradication therapies for *C. gattii* would be helpful and might inform optimal clinical management strategies for this potentially life-threatening pathogen.

## Supporting information

**S1 Data.**
(XLSX)

**S1 Fig. Consort diagram of the study.**
(TIF)

**S2 Fig. Site of infection for *C. gattii* and *C. neoformans* infection.**
(TIF)

**S1 Table. Imaging findings in patients with pulmonary disease.**
(DOCX)

**S2 Table. Imaging findings in patients with CNS disease.**
(DOCX)

**S3 Table. Symptoms of the patients and their vital signs at presentation, stratified by cryptococcal species.**
(DOCX)

**S4 Table. Laboratory findings of the patients at presentation stratified by cryptococcal species.**
(DOCX)

**S5 Table. Lumbar puncture findings stratified by cryptococcal species.**
(DOCX)

**S6 Table. Antifungal doses and duration stratified by cryptococcal species.**
(DOCX)

**S7 Table. Characteristics and clinical course of the patients with *Cryptococcus gattii* infection.**
(DOCX)

## Acknowledgments

The authors would like to acknowledge Mr Peter Horne for his assistance with the preparation of Figs 1 and 4.

## Author Contributions

**Conceptualization:** Beatrice Z. Sim, Luke Conway, Simon Smith, Josh Hanson.

**Data curation:** Beatrice Z. Sim, Luke Conway, Laura K. Smith, Lee Fairhead, Yi Shan Der, Lara Payne, Josh Hanson.

**Formal analysis:** Beatrice Z. Sim, Josh Hanson.

**Investigation:** Beatrice Z. Sim, Luke Conway, Simon Smith, Josh Hanson.

**Methodology:** Beatrice Z. Sim, Josh Hanson.

**Project administration:** Josh Hanson.

**Supervision:** Luke Conway, Simon Smith, Josh Hanson.

**Validation:** Josh Hanson.

**Visualization:** Josh Hanson.

**Writing – original draft:** Beatrice Z. Sim, Josh Hanson.

**Writing – review & editing:** Beatrice Z. Sim, Luke Conway, Laura K. Smith, Lee Fairhead, Yi Shan Der, Lara Payne, Enzo Binotto, Simon Smith, Josh Hanson.

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
