## [Decision Letter · Decision Letter 0]

21 Dec 2021

PONE-D-21-37969The aetiology and clinical characteristics of cryptococcal infections in Far North Queensland, tropical AustraliaPLOS ONE

Dear Dr. Hanson,

Thank you for submitting your manuscript to PLOS ONE. After careful consideration, we feel that it has merit but does not fully meet PLOS ONE’s publication criteria as it currently stands. Therefore, we invite you to submit a revised version of the manuscript that addresses the points raised during the review process. The manuscript has received a mixed review, and it requires significant revisions. Especially, serious are the concerns regarding the use of questionable statistics and lots of speculation.

If possible, deeper and more focused review of the data should be conducted. Whenever the data is insufficient to reach a viable conclusion the authors should acknowledge the need for more extensive studies with larger cohorts.  These concerns should be fully addressed.

We look forward to receiving your revised manuscript.

Kind regards,

Michal A Olszewski, DVM, PhD

Academic Editor

PLOS ONE

Journal Requirements:

3. We note that Figures 1 and 4 in your submission contain [map/satellite] images which may be copyrighted. All PLOS content is published under the Creative Commons Attribution License (CC BY 4.0), which means that the manuscript, images, and Supporting Information files will be freely available online, and any third party is permitted to access, download, copy, distribute, and use these materials in any way, even commercially, with proper attribution. For these reasons, we cannot publish previously copyrighted maps or satellite images created using proprietary data, such as Google software (Google Maps, Street View, and Earth). For more information, see our copyright guidelines: http://journals.plos.org/plosone/s/licenses-and-copyright.

a. You may seek permission from the original copyright holder of Figures 1 and 4 to publish the content specifically under the CC BY 4.0 license.  

4. Please include a copy of Table 2 which you refer to in your text on page 10.

Reviewers' comments:

Reviewer's Responses to Questions

**Comments to the Author**

1. Is the manuscript technically sound, and do the data support the conclusions?

Reviewer #1: Partly

Reviewer #2: Yes

2. Has the statistical analysis been performed appropriately and rigorously? 

Reviewer #1: Yes

Reviewer #2: Yes

3. Have the authors made all data underlying the findings in their manuscript fully available?

Reviewer #1: Yes

Reviewer #2: Yes

4. Is the manuscript presented in an intelligible fashion and written in standard English?

Reviewer #1: Yes

Reviewer #2: Yes

5. Review Comments to the Author

Reviewer #1: This paper reports the retrospective experience with 49 patients hospitalized in Northern Queensland with cryptococcosis. The report centers around 32 cases in which the isolate was identified by species: C neoformans or C. gattii. No novel findings were encountered. The higher prevalence of immunosuppressed patients with C. neoformans than C. gattii in Australia was confirmed. Some of the other difference between the two species in other reports were not found, in part due to the small sample size, such as more infections confined to the lung or more mass lesions in the brain among C. gattii cases.

The paper is burdened with masses of data that cannot be interpreted. The speculated connection between the higher prevalence of C. gattii cases in the last half of the year and flowering in Eucalyptus trees is specious. The lower, though normal, lymphocyte counts in the peripheral blood of C. neoformans patients is an example of applying a p value of 0.05 to more than 20 comparisons and finding one value of 0.04. There are larger and more detailed analyses of MRI and CT images, antifungal susceptibility and clinical descriptions of cryptococcosis in Australia than in this report. The long and speculative Discussion in the paper needs drastic shortening. The authors might consider deleting the data on the 17 cases with unspeciated isolates and minimizing speculation. If further analyses can provide insights into how cryptococcosis might differ in indigenous Australians, that might be instructive. Cairns has an experience with that population. Just because no difference was found doesn’t mean there were no differences.

Reviewer #2: In this manuscript, Sim et al. summarized the etiology, epidemiology and clinical course of cryptococcal infections in Far North Queensland between 1999 and 2019. During this period, infections with both Cryptococcus gattii and C. neoformans were identified. Individuals infected with C. gattii were less likely to be immunocompromised compared to patients with C. neoformans. Interestingly, they found that nearly 90% of the C. gattii infections had CNS involvement. Patients with C. gattii infections were more likely to be current or former smoker than patients with C. neoformans. Furthermore, more than 50% of the patients had a history of hazardous alcohol use. These clinical data are interesting; however, the manuscript will definitively benefit from an expanded mechanistic discussion on these clinical observations.

Comments

1) In this manuscript, 90% of the C. gattii infections had CNS involvement. Recent mouse studies showed that the primary target organ of C. gattii is different from that of C. neoformans. C. gattii grows faster in the lung and is primarily linked to pulmonary diseases (Pubmed PMID: 22570277). The authors should expand the discussion on this point.

2) Patients with C. gattii infections more likely to be smoker. It would be nice to have more discussion about that point.

3) It is also interesting that more than 50% of the patients had a history of hazardous alcohol use. The authors should discuss more on the potential mechanisms behind this observation. Actually, recent studies have shown that Liver cirrhosis is an independent risk factor of cryptococcal meningitis (Pubmed PMID: 25747471) and that liver macrophages play a prominent role to filter disseminating Cryptococcal yeast cells (Pubmed PMID: 31594939).

6. PLOS authors have the option to publish the peer review history of their article (what does this mean?). If published, this will include your full peer review and any attached files.

Reviewer #1: No

Reviewer #2: No

---

## [Author Response · Author response to Decision Letter 0]

9 Jan 2022

Response to reviewers

We thank the Editorial staff and the reviewers for the time that they have taken to review our manuscript and the very helpful comments that they have made to improve the work. Please find below our point-by-point responses to their comments. 

Editorial staff comments

Response: We have checked that the manuscript meets the style requirements - including file naming - and feel that it does. We are very happy to address specific concerns if any remain. 

Response: Requirement for informed consent was waived by the Ethics Committee as we highlight in lines 126-128. 

“The FNQ Human Research Ethics Committee provided ethical approval for the study (HREC/16/QCH/110 – 1083 LR). As the data were retrospective and de-identified, the Committee waived the requirement for informed consent”.

3. We note that Figures 1 and 4 in your submission contain [map/satellite] images which may be copyrighted. All PLOS content is published under the Creative Commons Attribution License (CC BY 4.0), which means that the manuscript, images, and Supporting Information files will be freely available online, and any third party is permitted to access, download, copy, distribute, and use these materials in any way, even commercially, with proper attribution. For these reasons, we cannot publish previously copyrighted maps or satellite images created using proprietary data, such as Google software (Google Maps, Street View, and Earth). For more information, see our copyright guidelines: http://journals.plos.org/plosone/s/licenses-and-copyright.

Response: All the maps in the submission were created using constructed using mapping software (MapInfo version 15.02, Connecticut, USA) using data provided by the State of Queensland (QSpatial) which are not copyrighted. Queensland Place Names —State of Queensland (Department of Natural Resources, Mines and Energy) 2019, available under Creative Commons Attribution 4.0 International licence https://creativecommons.org/licenses/by/4.0/ and ‘Coastline and state border–Queensland - State of Queensland (Department of Natural Resources, Mines and Energy) 2019, available under Creative Commons Attribution 4.0 International licence https://creativecommons.org/licenses/by/4.0/

These data - which are freely available in the public domain - are provided by the State Government of Queensland. This is a similar situation to some of the U.S. government sources that were suggested in the decision email. 

All the patients were managed in the Queensland State Government’s Public Health system. All the authors contributed to the manuscript when employed by the State Government of Queensland.

We have used these data to create maps in multiple PLoS publications in the past 4 years without any issue. 

1. PLoS Negl Trop Dis. 2021 Jun 21;15(6):e0009544. doi: 10.1371/journal.pntd.0009544. eCollection 2021 Jun.

2. PLoS Negl Trop Dis. 2021 Jan 14;15(1):e0008990. doi: 10.1371/journal.pntd.0008990. eCollection 2021 Jan.

3. PLoS One. 2020 Sep 3;15(9):e0238719. doi: 10.1371/journal.pone.0238719. eCollection 2020.

4. PLoS Negl Trop Dis. 2019 Jul 18;13(7):e0007583. doi: 10.1371/journal.pntd.0007583. eCollection 2019 Jul.

5. PLoS Negl Trop Dis. 2019 Feb 13;13(2):e0007205. doi: 10.1371/journal.pntd.0007205. eCollection 2019 Feb.

6. PLoS Negl Trop Dis. 2017 Mar 6;11(3):e0005411. doi: 10.1371/journal.pntd.0005411. eCollection 2017

We have added a footnote to Figures 1 and 4 - specifically addressing this copyright issue - to make this clearer.

4. Please include a copy of Table 2 which you refer to in your text on page 10.

Response: We thank the Editorial staff for highlighting this typographical error in which we had misnumbered the table in the original submission. This table is table 3 in the revised manuscript (page 13).

Reviewer comments

Reviewer #1: This paper reports the retrospective experience with 49 patients hospitalized in Northern Queensland with cryptococcosis. The report centers around 32 cases in which the isolate was identified by species: C neoformans or C. gattii. No novel findings were encountered. The higher prevalence of immunosuppressed patients with C. neoformans than C. gattii in Australia was confirmed. Some of the other difference between the two species in other reports were not found, in part due to the small sample size, such as more infections confined to the lung or more mass lesions in the brain among C. gattii cases.

The paper is burdened with masses of data that cannot be interpreted. The speculated connection between the higher prevalence of C. gattii cases in the last half of the year and flowering in Eucalyptus trees is specious. The lower, though normal, lymphocyte counts in the peripheral blood of C. neoformans patients is an example of applying a p value of 0.05 to more than 20 comparisons and finding one value of 0.04. There are larger and more detailed analyses of MRI and CT images, antifungal susceptibility and clinical descriptions of cryptococcosis in Australia than in this report. The long and speculative Discussion in the paper needs drastic shortening. The authors might consider deleting the data on the 17 cases with unspeciated isolates and minimizing speculation. If further analyses can provide insights into how cryptococcosis might differ in indigenous Australians, that might be instructive. Cairns has an experience with that population. Just because no difference was found doesn’t mean there were no differences.

Response: We thank the reviewer for their constructive comments. We agree that the discussion was too long and have shortened it significantly, by almost 30% (1217 to 864 words), as suggested.

We also agree with their comments about the limitations of the study which are largely a product of its sample size. Indeed, we highlight this specifically in lines 336-337. The risk of type 2 errors that we highlight in the limitations section (line 340) states precisely what the reviewer is suggesting. “Just because no difference was found doesn’t mean there were no differences” is the definition of a type 2 error.

Certainly, while there are other Australian analyses of larger size, this is the largest ever analysis in a tropical Australian location. The only comparable study - contained only 18 patients, only 12 of which were C. gattii - from the Northern Territory (Jenney et al. 2004, reference 7), was published over 15 years ago and examined patients managed between 1993 and 2000. The management of critical illness and cryptococcal disease has evolved since then. The significant difference in case-fatality rate between the two series highlights this and provides updated and contemporary data for the reader.

The fact that Indigenous Australians were over-represented in our cohort is also significant, and the reviewer’s comment regarding further analysis in this population is insightful. Further analyses have been presented as suggested (lines 147-156, lines 300-305 and table 2). 

The reviewer suggests that the paper is “burdened with masses of data” but it is important to note that most of this data is presented as supplementary material (7 supplementary tables, 2 supplementary figures and the dataset) which we present for the interested reader. There are only 3 tables of data in the revised submission which includes an additional table examining the burden of disease in Indigenous Australians as suggested by the reviewer.

The reviewer is correct to note that for every 20 analyses of a normal population, 1 - on average - will, by chance, have a p value of <0.05. However, we disagree with the reviewer when they state that the “lower, though normal, lymphocyte counts in the peripheral blood of C. neoformans patients is an example of applying a p value of 0.05 to more than 20 comparisons and finding one value of 0.04.” The median (IQR) lymphocyte count in patients with C. neoformans was 0.8 (0.5-1.3) x 106/L. The median lymphocyte count is below the normal reference range for the lymphocyte count in our laboratory (1.0-4.0 x 106/L), so the majority of patients with C. neoformans had lymphopenia. This is unlikely to be due to chance as 8/17 patients with confirmed C. neoformans had established immunocompromise (including 3 patients with advanced HIV - with lymphocyte counts of 0.1, 0.3 and 0.4 x 106/L respectively - and 2 patients with solid organ transplants with lymphocyte counts of 0.3 and 0.7 x 106/L respectively. This would accord with the knowledge that C. neoformans is a disease of immunodeficiency/immunocompromise and the lymphopenia is a marker of this, rather than a chance finding from multiple analyses.

We acknowledge the reviewer’s caution in suggesting correlation between observed data – it is important not to overstate our findings. Indeed, we specifically highlight this point in lines 344-345. We agree that it is important not to attribute causation between the flowering of the Eucalypt trees and cryptococcosis. Indeed, we are pains to make this point (lines 312-313). 

The association between Eucalyptus trees (specifically E. tereticornis and E. camaldulensis) is very well established (Ellis DH, Pfeiffer TJ J Clin Microbiol. 1990;28(7):1642., Sorrell TC, Chen SC, Ruma P, Meyer W, Pfeiffer TJ, Ellis DH, Brownlee AG J Clin Microbiol. 1996;34(5):1253.) so we would respectfully disagree with the reviewer who suggests that the analysis is “specious”. The data are presented only as hypothesis generating and would be, we believe, to be of interest to the readers of the article who would generally be only those with an interest in cryptococcal infection. In the large rural FNQ region there is an opportunity to look at a potential association between flowering of the trees that have been shown to harbour C. gattii; this is our attempt to convey these data to the readers.

Reviewer #2: In this manuscript, Sim et al. summarized the etiology, epidemiology and clinical course of cryptococcal infections in Far North Queensland between 1999 and 2019. During this period, infections with both Cryptococcus gattii and C. neoformans were identified. Individuals infected with C. gattii were less likely to be immunocompromised compared to patients with C. neoformans. Interestingly, they found that nearly 90% of the C. gattii infections had CNS involvement. Patients with C. gattii infections were more likely to be current or former smoker than patients with C. neoformans. Furthermore, more than 50% of the patients had a history of hazardous alcohol use. These clinical data are interesting; however, the manuscript will definitively benefit from an expanded mechanistic discussion on these clinical observations.

Comments

1) In this manuscript, 90% of the C. gattii infections had CNS involvement. Recent mouse studies showed that the primary target organ of C. gattii is different from that of C. neoformans. C. gattii grows faster in the lung and is primarily linked to pulmonary diseases (Pubmed PMID: 22570277). The authors should expand the discussion on this point.

Response: We thank the reviewer for the time that they have taken to review our paper and agree that there are several points that could be expanded in the discussion. We have done our best to do this while also attempting to shorten the discussion overall as per reviewer 1.

We have expanded our discussion about lung involvement particularly its association with alcohol and tobacco use (lines 297-302). We did not add the mouse model reference as we did not feel we had the space in the discussion to do justice to this point. It is also a little beyond the scope of the paper which focuses on human, clinical cases.

2) Patients with C. gattii infections more likely to be smoker. It would be nice to have more discussion about that point.

Response: We thank the reviewer for highlighting this point - we have expanded the discussion accordingly (lines 297-302). 

3) It is also interesting that more than 50% of the patients had a history of hazardous alcohol use. The authors should discuss more on the potential mechanisms behind this observation. Actually, recent studies have shown that Liver cirrhosis is an independent risk factor of cryptococcal meningitis (Pubmed PMID: 25747471) and that liver macrophages play a prominent role to filter disseminating Cryptococcal yeast cells (Pubmed PMID: 31594939).

Response: We agree that this is an interesting discussion point and we have expanded our discussion to include this (lines 297-302). We have added both of the suggested references for the interested reader.

---

## [Decision Letter · Decision Letter 1]

16 Feb 2022

PONE-D-21-37969R1The aetiology and clinical characteristics of cryptococcal infections in Far North Queensland, tropical AustraliaPLOS ONE

Dear Dr. Hanson,

Thank you for submitting your manuscript to PLOS ONE. After careful consideration, we feel that it has merit but does not fully meet PLOS ONE’s publication criteria as it currently stands. Therefore, we invite you to submit a revised version of the manuscript that addresses the points raised during the review process.

Both reviewers found the manuscript to improve significantly; however, there are a few issues that are unresolved per Reviewer 1, which can be corrected by editorial changes in the manuscript. Especially, please omit speculations and address other limitations that the reviewer 1 has identified. 

We look forward to receiving your final version of the revised manuscript, shortely.

Kind regards,

Michal A Olszewski, DVM, PhD

Academic Editor

PLOS ONE

Journal Requirements:

Reviewers' comments:

Reviewer's Responses to Questions

**Comments to the Author**

1. If the authors have adequately addressed your comments raised in a previous round of review and you feel that this manuscript is now acceptable for publication, you may indicate that here to bypass the “Comments to the Author” section, enter your conflict of interest statement in the “Confidential to Editor” section, and submit your "Accept" recommendation.

Reviewer #1: (No Response)

Reviewer #2: All comments have been addressed

2. Is the manuscript technically sound, and do the data support the conclusions?

Reviewer #1: Partly

Reviewer #2: Yes

3. Has the statistical analysis been performed appropriately and rigorously? 

Reviewer #1: Yes

Reviewer #2: I Don't Know

4. Have the authors made all data underlying the findings in their manuscript fully available?

Reviewer #1: Yes

Reviewer #2: Yes

5. Is the manuscript presented in an intelligible fashion and written in standard English?

Reviewer #1: Yes

Reviewer #2: Yes

6. Review Comments to the Author

Reviewer #1: This is better manuscript. However, the following issues should be addressed.

This is a report of 49 patients with cryptococcosis, of whom 16 may have had infection confined to the lung (table 2). Most analyses were done on a subset of 32 patients in whom the Cryptococcus was speciated as neoformans or gattii. Of the 32, 4 had isolated pulmonary disease. Though the diagnosis, management and outcome differs for cryptococcosis confined to the lung, that distinction does not appear in analysis of management or outcome. This should be corrected.

25.change wording to “clinical course of 49 cryptococcal infections”. Please insert total number of case so number of 32 in next line is in perspective

29-30 . omit speculation about flowering of trees because this new speculation does not have sufficient confirmation in the paper and may mislead readers (see below)

31 and 52-53 wording suggests that cryptococcomas may benefit by neurosurgery, presumably meaning resection. This is not true and is misleading.

93 a map of Australia is readily available to readers outside this manuscript and is unnecessary

138. Chart reviews about smoking and alcohol use find either no mention, simple mention or quantitation (pack years of cigarettes, amount of alcohol). Lines 109-110 indicate that “no mention”’ was tallied as no use. Cultural expectations may factor into that question. For example, was an indigenous patient more likely to be asked? The authors should be cautious in chart review interpretations, such as smokers being more common.

141 table 1 would benefit by a column with the results of the 17 patients with an unspeciated isolate. Readers don’t know how much bias is introduced by omitting a third of the cohort

171-172 This reference to the season of diagnosis correlating with tree flowering has several important limitations that are not adequately taken into account when the authors put this speculation in the abstract. As pointed out in lines 311-312 the time from exposure to clinical infection with C. gattii is unknown. However, it is thought to be from months to many years. This tentative conclusion is based on finding cases of C. gattii in patients who have left known endemic areas years previously. A long incubation period would help explain why there are no case clusters. Even in the Vancouver Island outbreak, there was no temporal relationship between time on the island and onset of symptoms. C. gattii has been isolated from dozens of tree species around the world but exposure history to such trees has not been correlated with disease, a similar situation with C. neoformans and weathered pigeon droppings. Nor has anyone correlated exposure to flowering of any tree nor flowering to isolation of the fungus from the tree. The authors should not put in the abstract (lines 29-30) their speculation that exposure to a flowering eucalyptus tree might lead to C. gattii infection.

Fig.3 There are many possible reasons for differing seasons at diagnosis but recent exposure is not the most likely.

Fig. 4 A similar reservation exists about geographic location. The location of a patient at time of diagnosis depends on population mobility. For example, an indigenous population may be less mobile. Location of C. gattii cases in rural areas may be a function of mobility.

Fig 6 and table 3 include drugs not recommended for treatment and don’t contribute to any conclusions.

Reviewer #2: (No Response)

7. PLOS authors have the option to publish the peer review history of their article (what does this mean?). If published, this will include your full peer review and any attached files.

Reviewer #1: No

Reviewer #2: No

---

## [Author Response · Author response to Decision Letter 1]

7 Mar 2022

Response to reviewers

We thank the Editorial staff and the reviewers for the time that they have taken to review our manuscript and the very helpful comments that they have made to improve the work. Reviewer 2 appears happy with the revision, however Reviewer 1 still has some valid concerns. Please find below our point-by-point responses to Reviewer 1’s comments. 

Reviewers’ comments

Reviewer 1

This is a report of 49 patients with cryptococcosis, of whom 16 may have had infection confined to the lung (table 2). Most analyses were done on a subset of 32 patients in whom the Cryptococcus was speciated as neoformans or gattii. Of the 32, 4 had isolated pulmonary disease. Though the diagnosis, management and outcome differs for cryptococcosis confined to the lung, that distinction does not appear in analysis of management or outcome. This should be corrected.

Response: We thank the reviewer for highlighting this important point. In fact, one of the patients with C. neoformans infection had disease limited to the skin, an oversight we have now corrected (lines 203-204). The reviewer is right to note that patients without CNS disease have different management and a different prognosis to patients with CNS involvement. We have added this as a limitation of the study (lines 350-353). However, as there were only 4/32 (12.5%) in the cohort with a speciated organism without CNS involvement, this is unlikely to have affected our findings significantly. Furthermore, there was no statistical difference in the case fatality rate in this series in patients with and without CNS involvement in the cohort, data that we now present in lines 277-278. 

25.change wording to “clinical course of 49 cryptococcal infections”. Please insert total number of case so number of 32 in next line is in perspective

Response: We thank the reviewer for highlighting this point. The text has been corrected accordingly (lines 25-27)

29-30 . omit speculation about flowering of trees because this new speculation does not have sufficient confirmation in the paper and may mislead readers (see below)

Response: This has been removed from the abstract as requested.

31 and 52-53 wording suggests that cryptococcomas may benefit by neurosurgery, presumably meaning resection. This is not true and is misleading.

Response: The neurosurgical management of cryptococcomas that we were describing was, in fact, management of elevated intracranial pressure using ventriculoperitoneal shunting or external ventricular drainage, although resection is also reported in the literature. We have added 2 references - Chan et al. Neurosurgical aspects of cerebral cryptococcosis. Neurosurgery 1989, and Chastain et al. Cerebral Cryptococcomas: A Systematic Scoping Review of Available Evidence to Facilitate Diagnosis and Treatment. Pathogens 2022 - which examine the neurosurgical management of cryptococcomas (references 4 & 5).

93 a map of Australia is readily available to readers outside this manuscript and is unnecessary

Response: The reviewer is right to note that most readers would be aware of the country of Australia. However, we included the map to highlight the tropical - rather than temperate - location of our study and distinguishing it from other published tropical Northern Australia cohorts e.g., the Northern Territory (reference 9). But we would be happy for the Editors to remove the map if they felt that it added little to the paper.

138. Chart reviews about smoking and alcohol use find either no mention, simple mention or quantitation (pack years of cigarettes, amount of alcohol). Lines 109-110 indicate that “no mention”’ was tallied as no use. Cultural expectations may factor into that question. For example, was an indigenous patient more likely to be asked? The authors should be cautious in chart review interpretations, such as smokers being more common.

Response: We thank the reviewer for highlighting this point. We agree that the data collection was limited by its retrospective collection, a point that we acknowledge in the discussion (lines 348-349). 

141 table 1 would benefit by a column with the results of the 17 patients with an unspeciated isolate. Readers don’t know how much bias is introduced by omitting a third of the cohort

Response: We thank the reviewer for highlighting this point. We have added a column as suggested by the reviewer.

171-172 This reference to the season of diagnosis correlating with tree flowering has several important limitations that are not adequately taken into account when the authors put this speculation in the abstract. As pointed out in lines 311-312 the time from exposure to clinical infection with C. gattii is unknown. However, it is thought to be from months to many years. This tentative conclusion is based on finding cases of C. gattii in patients who have left known endemic areas years previously. A long incubation period would help explain why there are no case clusters. Even in the Vancouver Island outbreak, there was no temporal relationship between time on the island and onset of symptoms. C. gattii has been isolated from dozens of tree species around the world but exposure history to such trees has not been correlated with disease, a similar situation with C. neoformans and weathered pigeon droppings. Nor has anyone correlated exposure to flowering of any tree nor flowering to isolation of the fungus from the tree. The authors should not put in the abstract (lines 29-30) their speculation that exposure to a flowering eucalyptus tree might lead to C. gattii infection.

Response: It is clear that the reviewer is sceptical about the validity of the potential association between the flowering of Eucalyptus trees and Cryptococcus gattii disease. However, it is not quite correct to state that there is no-one has correlated flowering of any tree and isolation of Cryptococcus gattii. Australian experts in the field have previously presented the correlation between flowering of the Eucalyptus trees and airborne dispersal of C. gattii basidiospores that have the potential to act as infectious propagules (Ellis DH, Pfeiffer TJ J Clin Microbiol. 1990;28(7):1642-1644). In this paper, Ellis and Pfeiffer reported the results of an 8-month sampling period - during which 2100 collections were made - and describe a correlation between isolation of airborne basidiospores and flowering of the E. camaldulensis in late spring. These basidiospores were present in the environment for only short periods of time. When the air sampling was performed under other trees and under E. camaldulensis trees not in flower, no C. gattii was detected. 

On Vancouver Island, C. gattii air concentrations were also noted to be higher in the summer months (Kidd et al. Emerg Infect Dis. 2007;13(1):51), although this was not linked to the pollination or flowering of any specific trees and the airborne C. gattii was identified year-round in this series. However, none of the trees that were examined in this series were Eucalypts and the dominant C. gattii genotypes in the two regions are different (VGI in Australia versus VGII in British Columbia). It may be that the different climates and flora of Australia and North America might lead to differences in the ecology of C. gattii in the two regions. 

We agree with the reviewer that multiple factors are almost certainly involved in the pathogenesis of the disease and in this revised manuscript we present the limitations of our hypothesis in more detail and propose other factors that might also contribute to the local incidence of disease (lines 321-330). In the interest of not ascribing too much significance to the observation, we have removed the any reference to flowering from the abstract, as requested by the reviewer. 

However, the identification of airborne infectious propagules in the aforementioned Australian and Canadian reports would suggest that this is one way that the organism can reach the patients and establish infection in the lung and beyond. There is an association in this and other Australian series between rural residence and C. gattii disease (Jenney et al. JCM 2004, Chen et al CID 2012), and in the large rural FNQ region there is an opportunity to look at a potential association between flowering of the trees that have been shown to harbour C. gattii. This is our attempt to convey these data to the readers. Hopefully the Editor feels that in this revised version we have presented both sides of the argument in a balanced - and less speculative - manner.

Fig.3 There are many possible reasons for differing seasons at diagnosis but recent exposure is not the most likely.

Response: We agree with the reviewer and have revised the manuscript to present these factors - in, we believe, a balanced manner - for the reader (lines 313-330).

Fig. 4 A similar reservation exists about geographic location. The location of a patient at time of diagnosis depends on population mobility. For example, an indigenous population may be less mobile. Location of C. gattii cases in rural areas may be a function of mobility.

Response: We agree with the reviewer; this is a limitation of this retrospective study that we now specifically highlight in the discussion (lines 349-350).

Fig 6 and table 3 include drugs not recommended for treatment and don’t contribute to any conclusions.

Response: It is true that amphotericin, fluconazole and flucytosine are the agents used most commonly in the management of Cryptococcal disease. However, the 2010 IDSA guidelines for the management of Cryptococcal disease suggest that itraconazole, voriconazole and posaconazole are acceptable second line agents if first line therapies in are not tolerated or contraindicated (Perfect et al. Clin Infect Dis 2010). 

Other studies reporting the antifungal susceptibilities of C. gattii present data on susceptibility to a variety of agents. This is interesting from a clinical perspective as second line therapies may be required due to drug intolerance or potential drug-drug interactions. It is also interesting from a microbiological perspective and speaks to the global generalisability of treatment recommendations. 

One study from Spain, for instance, reported that fluconazole, voriconazole, and posaconazole MICs were significantly higher in C. gattii than C. neoformans (Torres-Rodríguez et al. J Antimicrob Chemother. 2008 Jul;62(1):205-6). While another from Brazil - which examined the susceptibility of 57 strains of C. gattii to nine antifungal agents - reported significantly higher MICs for fluconazole, voriconazole, amphotericin B, and flucytosine in C. gattii than in C. neoformans (Trilles et al. J Clin Microbiol. 2004;42(10):4815.). A Chinese paper published only last month (Zang et al. PLoS NTD. 2022) reported the susceptibility to 6 antifungal agents (including voriconazole, itraconazole and posaconazole) 

We have therefore included extended susceptibilities for the interested reader.

---

## [Editor Report · Decision Letter 2]

8 Mar 2022

The aetiology and clinical characteristics of cryptococcal infections in Far North Queensland, tropical Australia

PONE-D-21-37969R2

Dear Dr. Hanson,

We’re pleased to inform you that your manuscript has been judged scientifically suitable for publication and will be formally accepted for publication once it meets all outstanding technical requirements.

Kind regards,

Michal A Olszewski, DVM, PhD

Academic Editor

PLOS ONE

---

## [Editor Report · Acceptance letter]

22 Mar 2022

PONE-D-21-37969R2 

The aetiology and clinical characteristics of cryptococcal infections in Far North Queensland, tropical Australia 

Dear Dr. Hanson:

I'm pleased to inform you that your manuscript has been deemed suitable for publication in PLOS ONE. Congratulations! Your manuscript is now with our production department. 

Kind regards, 

on behalf of

Dr. Michal A Olszewski 

Academic Editor

PLOS ONE